# Fabrication of Antibacterial Sponge Microneedles for Sampling Skin Interstitial Fluid

**DOI:** 10.3390/pharmaceutics15061730

**Published:** 2023-06-14

**Authors:** Jianmin Chen, Xiaozhen Cai, Wenqin Zhang, Danhong Zhu, Zhipeng Ruan, Nan Jin

**Affiliations:** 1School of Pharmacy, Fujian Medical University, Putian 351100, China; 2Key Laboratory of Pharmaceutical Analysis and Laboratory Medicine, Putian University, Putian 351100, China; 3School of Pharmacy and Medical technology, Putian University, Putian 351100, China

**Keywords:** microneedles, antibacterial activity, silver nanoparticles, ISF sampling, medical devices

## Abstract

Microneedles (MNs) have recently garnered extensive interest concerning direct interstitial fluid (ISF) extraction or their integration into medical devices for continuous biomarker monitoring, owing to their advantages of painlessness, minimal invasiveness, and ease of use. However, micropores created by MN insertion may provide pathways for bacterial infiltration into the skin, causing local or systemic infection, especially with long-term in situ monitoring. To address this, we developed a novel antibacterial sponge MNs (SMNs@PDA-AgNPs) by depositing silver nanoparticles (AgNPs) on polydopamine (PDA)-coated SMNs. The physicochemical properties of SMNs@PDA-AgNPs were characterized regarding morphology, composition, mechanical strength, and liquid absorption capacity. The antibacterial effects were evaluated and optimized through agar diffusion assays in vitro. Wound healing and bacterial inhibition were further examined in vivo during MN application. Finally, the ISF sampling ability and biosafety of SMNs@PDA-AgNPs were assessed in vivo. The results demonstrate that antibacterial SMNs enable direct ISF extraction while preventing infection risks. SMNs@PDA-AgNPs could potentially be used for direct sampling or combined with medical devices for real-time diagnosis and management of chronic diseases.

## 1. Introduction

Microneedles (MNs) are micro-scale projections (25–2500 μm in size) that were originally designed to breach the skin barrier for enhanced drug delivery [1]. The concept of MNs first emerged in the 1970s, but the first study on MNs for transdermal drug delivery was not reported until the late 1990s [2]. Owing to their numerous benefits, such as painlessness, minimal invasiveness, ease of use, avoidance of first-pass metabolism, and improved drug efficacy [3], MNs have attracted enormous interest over the past two decades and have been widely used for drug delivery, especially of biomacromolecules such as genes, peptides, proteins, and vaccines [4,5]. In addition to drug delivery, MNs have been integrated into medical devices for biosampling, diagnosis, and biosensing [6,7]. 

In recent years, there has been increasing interest in developing MNs-based medical devices for interstitial fluid (ISF) sampling, disease diagnosis, therapeutic drug monitoring, and even real-time monitoring of chronic diseases [8]. As a crucial component of such devices, MNs should be inserted into epithelial tissues (mainly the epidermis or intestinal epithelium) to act as a conduit between the body and the device [9]. Although MNs are minimally invasive, the hundreds of micropores created in the stratum corneum by MNs may also pose risks of skin infection [10], especially for long-term and real-time monitoring of chronic diseases (e.g., diabetes and cancer). In fact, many studies have focused on real-time monitoring of blood glucose levels in diabetes patients, who are more prone to local bacterial infection [11]. Therefore, it is imperative to develop MNs with antibacterial activity to mitigate the risk of infection. 

A facile method to endow MNs with antibacterial properties is to coat a layer of antibacterial agent film on their surface. Hundreds of antimicrobial agents, including various natural or synthetic organic and inorganic substances, have been developed [12]. Among them, silver nanoparticles (AgNPs) have strong antibacterial effects against both Gram negative and positive bacteria [13]. In addition, AgNPs can circumvent antibiotic resistance due to their lower resistance to drugs [14]. As a result, AgNPs have been widely used in antibacterial medical products, including wound dressings, catheters, and bandages [15,16,17]. In addition to various traditional methods for synthesizing AgNPs, many green strategies have been proposed and rapidly developed because they are simple, reliable, environmentally friendly, cost-effective, and easy to scale up [18]. Among these strategies, the deposition of AgNPs on polydopamine (PDA) coatings is promising, due to its excellent properties such as good biocompatibility, strong adhesion, and abundant chemical functionalities [19]. A large number of studies have shown that the deposition of AgNPs on PDA coatings could endow the coating on various substrates with antibacterial properties [20]. 

Our previous study proposed a novel type of MNs, called sponge MNs (SMNs), for rapid ISF sampling [21]. SMNs show potential as an essential component of medical devices for real-time monitoring of chronic diseases. However, inserting MNs into the skin may risk infection. To address this, we designed and developed the first generation of antibacterial SMNs by coating AgNPs onto PDA-modified SMNs (SMNs@PDA-AgNPs). To evaluate the properties and performance of SMNs@PDA-AgNPs, we conducted a series of analyses and tests. The morphology and liquid extraction capacity were examined to confirm the successful PDA-AgNPs modification and ISF sampling ability. The mechanical strength was measured to ensure the penetration of MNs into the skin. The antibacterial activity was evaluated both in vitro and in vivo to demonstrate the long-lasting effects of AgNPs. The analyses of SMNs@PDA-AgNPs suggest their promise as platforms for ISF extraction with minimal infection risks. They could potentially couple with or embed into medical devices to diagnose and manage chronic diseases in real time.

## 2. Materials and Methods

### 2.1. Materials 

Polyvinyl alcohol (PVA), formaldehyde, sulfuric acid, starch, n-pentane, 1,4-dioxane, trihydroxymethyl-aminomethane (Tris) and hydrochloric acid (HCl) were obtained from Aladdin Bio-technology (Shanghai, China). Dopamine hydrochloride (DA), silver nitrate (AgNO_3_), methylene blue, glucose assay kit (GAGO20), and cholesterol quantification kit (MAK043) were acquired from Sigma-Aldrich (St. Louis, MO, USA). *Escherichia coli* (*E. coli*) was purchased from Fu Xiang Biological Technology (Shanghai, China). The blood glucose meter (Nipro Diagnostics Inc., Coral Gables, FL, USA) and cholesterol meter (BeneCheck, Taiwan) were procured from a local drugstore (Putian, China). All other chemicals used were of analytical or pharmaceutical grade.

### 2.2. Animals

Male Sprague Dawley rats (approximately 200 g) were obtained from Wu Shi Laboratory Animal Company (Fuzhou, China) and housed under specific pathogen-free conditions (22 ± 1 °C, 40–60% relative humidity). The animals were allowed one week to acclimate before experiments. All animals were treated in accordance with protocols (202030) approved by the Institutional Animal Care and Use Committee, Putian University, following the international ethical guidelines for animal research.

### 2.3. Fabrication of SMNs

SMNs were prepared using the solution casting method previously reported by our group [21]. Briefly, a mixture solution of 0.1 g starch, 3 mL PVA solution (12.5 wt%), 0.4 mL sulfuric acid (50 wt%), 0.5 mL formaldehyde solution (37 wt%), and 0.15 mL n-pentane was prepared and homogeneously mixed by stirring for 2 min. Subsequently, the mixture was added to a PDMS mold containing 144 inverted micropores arranged in a 12 × 12 array on the surface. The mold was then placed in a high pressure (approximately 3 atm) cylinder for 5 min to allow the solution to enter the micropores. After that, the mold was placed in a drying oven (40 °C) for 3 h for polymerization to form SMNs. Finally, the SMNs were peeled from the mold, rinsed with water, and dried at 40 °C for 2 h.

### 2.4. Preparation of SMNs@PDA-AgNPs

SMNs@PDA-AgNPs were prepared using the in situ reduction method reported in the literature [22]. In brief, DA powder was dissolved in Tris buffer (pH 8.5, 10 mM) to prepare DA solutions with different concentrations (0.1, 0.2, 0.4, 0.8, and 1.5 mg/mL). SMNs were immersed directly in freshly prepared DA solutions at room temperature (25 °C) for various durations (1, 2, 4, 8, and 16 h) to form a PDA coating by the self-polymerization of DA. The obtained SMNs@PDA were rinsed with water and dried under N_2_ gas flow. Subsequently, they were immersed in AgNO_3_ solutions with different concentrations (0.25, 0.5, 1, 2, and 4 mM) at room temperature (25 °C) for different reaction time periods (2, 4, 8, 16, and 24 h) to grow AgNPs on the surface of SMNs@PDA. Finally, the prepared SMNs@PDA-AgNPs were washed with water three times and dried at 40 °C for 6 h. For comparison, SMNs were treated using the same procedures mentioned above without being dipped into DA solutions, and the samples obtained were denoted as SMNs-AgNO_3_.

### 2.5. Characterization of SMNs@PDA-AgNPs

The macroscopic morphologies of SMNs, SMNs@PDA, SMNs-AgNO_3_, and SMNs@PDA-AgNPs were examined using a camera (Canon EOS 700 D, Kyushu, Japan) equipped with a macro lens and optical microscope. The microscopic morphologies of these MNs were characterized by scanning electron microscopy (SEM, Zeiss Sigma 300, Jena, Germany) at an accelerating voltage of 5 kV. Energy dispersive X-ray spectroscopy (EDS) was performed using the X-Max 80 Silicon Drift Detector (Oxford Instruments, Oxford, UK) in SEM to analyze the chemical elements present on the surfaces of these samples. In addition, the chemical compositions of these samples were examined by X-ray diffraction (XRD, Rigaku 2500PC, Tokyo, Japan) using Cu Kα radiation in continuous scanning mode (40 kV, 40 mA and λ = 1.54056 Å).

### 2.6. Evaluation of Liquid Extraction Capacity 

The liquid extraction capacity of different MNs was determined using the previously reported method [23]. Briefly, the dry weight of MNs (W_0_) was weighed before testing. Thereafter, a simple artificial skin model was constructed by coating 1.0 wt% agar hydrogel (mimicking the dermis with ISF) with a Parafilm (Heathrow Scientific, Vernon Hills, IL, USA) membrane, which was used to represent the water-impermeable stratum corneum and epidermis. Subsequently, the MNs were carefully inserted into the skin model, allowing the MNs to begin extracting liquid from the hydrogel layer. At pre-determined time points (1, 5, 10, 20, 40, 60, and 120 min), the MNs were taken out and the sample weight (W_t_) was weighed immediately. Three independent samples were tested (*n* = 3) and the swelling ratios were calculated from the following equation: swelling ratio (%) = (W_t_ − W_0_)/W_0_ × 100%. In addition, the weight of the liquid extracted by MNs was calculated directly by W_t_ minus W_0_. 

### 2.7. Mechanical Test and In Vitro Insertion Study

To ensure that SMNs@PDA-AgNPs have sufficient strength to penetrate through the skin without fracture, a force gauge (Handpi NK-50, Shenzhen, China) was applied to study the mechanical properties of MNs as described in our previous study [24]. Briefly, the MNs were attached to a horizontal platform with the needle tips facing upwards. A probe was moved downwards perpendicularly towards the MNs at a constant speed of 0.5 mm/min. Both the applied force and the morphological changes in the MNs were continuously recorded. When the probe moved down by 0.1 mm, the applied force was considered as the maximum bearing force of the MNs. Therefore, the maximum bearing force of each tip (N/tip) could be calculated from the value of the force divided by the number of tips.

To evaluate the skin penetration capacity of SMNs@PDA-AgNPs, aluminum foil was used as an artificial skin simulant according to the reported method with some modifications [25]. The SMNs@PDA-AgNPs were manually inserted into the aluminum foil for 1 min to form hundreds of micropores in the foil, through which a trypan blue solution was applied to white paper. In addition, the SMNs@PDA-AgNPs were applied manually to excised rat skin for 3 min to investigate the skin penetration capacity. After removal of the SMNs@PDA-AgNPs, the treated skin was stained with trypan blue solution and then photographed using a camera. The penetration efficiency (PE) can be calculated from the equation: PE (%) = (number of stained micropores/number of tips) × 100%. Moreover, the treated skin was fixed in paraformaldehyde (4%), dehydrated, embedded in paraffin, and sectioned, followed by Hematoxylin and Eosin (H&E) staining.

### 2.8. Assessment of Antibacterial Activity In Vitro

The antibacterial activities of these MNs were tested using the agar disc diffusion method with some adjustments [26]. To obtain a round zone of bacterial inhibition (ZOBI), the MNs were trimmed to a 10 mm diameter disc, followed by UV disinfection for 30 min. A 100 μL of *Eacherichia coli* (*E. coli*, 1 × 10^5^ CFU/mL) suspension was pipetted and evenly coated on Petri dish (~90 mm diameter) containing Lysogeny Broth (LB) agar plates. After that, the MNs were gently inserted into the agar plate by hand. Following 24 h of incubation at 37 °C, the ZOBI was photographed. The ZOBI diameter was determined using image analysis software (Image-Pro Plus version 6.0). To optimize the antibacterial performance of the SMNs@PDA-AgNPs, the most important factors affecting the antibacterial activities, including the concentration and incubation time of both the DA and AgNO_3_ solutions, were further investigated using the same procedures mentioned above.

### 2.9. Evaluation of Antibacterial Activity In Vivo 

To verify the antibacterial activity of SMNs@PDA-AgNPs in vivo, a wound healing experiment was carried out according to the reported method [27]. For comparison, SMNs were immersed in a ciprofloxacin (CIP, 1 mg/mL) solution for 1 h and then dried in a vacuum oven (40 °C); these were denoted as SMNs + CIP. Fifteen rats were randomly divided into three groups (*n* = 5): (i) SMNs (negative control); (ii) SMNs@PDA-AgNPs; and (iii) SMNs + CIP (positive control). Two days before the experiment (−2 day), a circular wound with a 20 mm diameter was created on the back of each rat and photographed. A 100 μL *E. coli* suspension (1 × 10^5^ CFU/mL) was pipetted onto the wound area and incubated for 2 days to develop biofilms. The infected rats were kept individually in cages with plenty of food and water. After that, the rats were treated according to the different grouping designs. The wounds were photographed and measured at 0, 2, 4, 6, 10, and 14 days after treatment. The wound healing rate (%) = (Initial wound area – Residual wound area on different days)/Initial wound area × 100%. After 14 days, the experiment was completed and the rats were sacrificed. 

### 2.10. Evaluation of Antibacterial Activity for Long-Term Application

The SMNs@PDA-AgNPs were designed as a component of medical devices rather than a wound healing device. Therefore, the antibacterial activity of the MNs during practical application was further investigated. Fifteen rats were randomly divided into three groups (*n* = 5): (i) SMNs without bacterial infection; (ii) SMNs@PDA-AgNPs with bacterial infection; and (iii) SMNs with bacterial infection. The hair on the back of each rat was removed to develop an experiment area with a 20 mm diameter. A 100 μL *E. coli* suspension (1 × 10^5^ CFU/mL) was pipetted onto the area, followed by treatment of group ii and iii. In group (i), normal saline was given instead of *E. coli*. After applying the MNs, a sterile bandage was used to secure the MNs and cover the entire skin area for experiment. After 2 days, the MNs were removed from the rats. One rat from each group was sacrificed and the treated skin was cut off, followed by H&E staining. The other rats were kept individually in cages for further observation. Throughout the experiment, the skin was photographed and transepidermal water loss (TEWL) was monitored using our reported method [28]. 

### 2.11. Application of MNs in Sampling ISF In Vivo 

The sampling performance of SMNs@PDA-AgNPs was evaluated in vivo using our reported method [21]. Briefly, the weight of SMNs@PDA-AgNPs (W_0_) was measured, and they were applied to the back skin of a rat. After 20 min, the weight of the MNs (W_t_) containing ISF was immediately measured. The MNs were then transferred to a centrifuge tube with 100 μL of a solvent mixture (ethanol: water = 1:1) to recover the extracted glucose and cholesterol. Following centrifugation at 5000 rpm for 10 min, the fluid was collected. The concentrations (C_t_) of glucose and cholesterol in the fluid were tested using glucose assay kit and cholesterol quantification kit, respectively. Therefore, the concentration (C_s_) of glucose and cholesterol in the ISF could be calculated according to the following equation:$$\;C_{s}=\frac{C_{t}\times \;V}{\left(W_{t\;}-{\;W}_{0}\right)\;/\;\rho },$$
where ρ (approximately equal to 1 g/mL) represents the ISF density. Five rats were used in this experiment, and ISF was collected from different skin sites of each rat three times. For comparison, blood glucose and cholesterol concentrations (collected from the tail vein) were detected using a glucose meter (Nipro Diagnostics Inc., Coral Gables, FL, USA) and a cholesterol meter (BeneCheck, Taipei, Taiwan), respectively. In addition, an experiment was conducted to study the recovery process of the micropores induced by treating with the SMNs@PDA-AgNPs for 20 min. The treated skin was photographed, and TEWL was measured at predetermined time points (0, 0.5, 1, 2, 4, 6, 8, and 10 h).

### 2.12. Statistical Analysis

All experiments were performed in triplicate at a minimum. The data were collected and processed using one-way analysis of variance in Microsoft Excel^®^ 2010 (Microsoft Corporation, Redmond, DC, USA). The data are presented as the mean ± standard deviation (SD). * *p* < 0.05, ** *p* < 0.01 and *** *p* < 0.001 were considered statistically significant.

## 3. Results and Discussion

### 3.1. Fabrication and Macromorphology of MNs

SMNs have been designed and developed as components of medical devices for ISF sampling by our group previously [21]. However, the long-term application of SMNs on the skin may increase the risk of bacterial infection. Therefore, antibacterial SMNs@PDA-AgNPs were designed and prepared by depositing AgNPs on the PDA surface of SMNs. The fabrication process and application of the SMNs@PDA-AgNPs are presented in Figure 1. We hypothesized that the SMNs@PDA-AgNPs could penetrate the skin efficiently and exhibit antibacterial properties during application, and thus may have significant potential in medical device applications.

Photographs of SMNs, SMNs@PDA, SMNs-AgNO_3_, and SMNs@PDA-AgNPs are shown in Figure 2a(i–iv), respectively. It can be observed that all types of MNs consist of a 12 × 12 array in an area of 10 mm by 10 mm. The macro-morphologies of these MNs show no obvious differences, indicating that the fabrication procedures of SMNs@PDA-AgNPs did not significantly affect the basic structure of the MNs. Specifically, the SMNs were white in color with a foam-like structure, while the SMNs@PDA appeared dark brown with the same structure. The difference in color is due to the coating of the PDA film, which is a dark brown polymer obtained by the self-polymerization of DA [29]. The SMNs-AgNO_3_ have the same color as SMNs, so they were stained with rhodamine B to distinguish them from the others. The color of SMNs@PDA-AgNPs was dark black due to the deposition of AgNPs on the surface, which were black and synthesized through in situ reduction by PDA [30]. The mechanism of PDA-assisted synthesis of AgNPs can be summarized as follows [22]: Dopamine undergoes a series of reactions, ultimately resulting in the formation of PDA. PDA, characterized by active phenolic hydroxyl groups, exhibits the ability to effectively react with silver ions, leading to their reduction and subsequent synthesis of AgNPs. Moreover, as shown in Figure 2b(i–iv), the pyramid structure and sharpness of the needle tips of these MNs are all well maintained, and there are no significant differences among them.

### 3.2. Liquid Extraction Capacity of MNs

SMNs@PDA-AgNPs have been developed to sample ISF; therefore, a robust liquid extraction capability is highly desirable to ensure that they can rapidly extract liquid from tissues. Although SMNs have been reported to have excellent liquid extraction capacity in our previous study, it is unclear whether the deposition of AgNPs will affect the liquid extraction capacity of SMNs@PDA-AgNPs. Therefore, the liquid extraction capacities of different MNs were tested, and the results are presented in Figure 2c,d. 

The swelling ratios of SMNs, SMNs@PDA, SMNs-AgNO_3_, and SMNs@PDA-AgNPs were 13.04%, 2.87%, 7.82%, and 1.31% at the first minute, respectively, as shown in Figure 2c. As time increased, the swelling ratios of all MNs increased. It is evident that the swelling ratio and speed of SMNs@PDA-AgNPs are significantly lower than those of the other MNs. Specifically, the swelling ratios of SMNs, SMNs@PDA, and SMNs-AgNO_3_ at 60 min were 207.94%, 198.31%, and 194.26%, which were 1.96, 1.87, and 1.83 times higher than that of SMNs@PDA-AgNPs (106.20%). The results showed that depositing AgNPs would reduce the liquid extraction capacity and speed of the MNs, consistent with the results of a previous study [31]. This can be attributed to the deposition of AgNPs blocking the porous structure of the MNs and shrinking the pore volume for liquid absorption.

In addition, the amount of liquid extracted from these MNs is shown in Figure 2d. Similarly, the extracted liquid amounts of SMNs@PDA-AgNPs were significantly less than those of the other MNs at each determined time point. However, 10.93 mg of extracted liquid was obtained with SMNs@PDA-AgNPs at 60 min, far superior to the other study (0.84 mg) [32]. Although the incorporation of AgNPs slightly compromised the liquid extraction capacity, SMNs@PDA-AgNPs still have an excellent ability to extract ISF rapidly.

### 3.3. Mechanical Properties and Skin Penetration Capacity of MNs

The maximum bearing force per tip (N/tip) of the different MNs was determined to ensure that the MNs have sufficient strength to penetrate the skin without breaking. As shown in Figure 2e, the forces of SMNs, SMNs@PDA, SMNs-AgNO_3_, and SMNs@PDA-AgNPs are 0.183 ± 0.013 N, 0.172 ± 0.022 N, 0.197 ± 0.017 N, and 0.180 ± 0.014 N, respectively. These are all greater than the minimum average force required for normal skin penetration (0.045 N) [33]. The results show that SMNs@PDA-AgNPs have sufficient strength to penetrate the skin without breaking. In addition, there was no significant difference (*p* < 0.05) in force between the SMNs and SMNs@PDA-AgNPs, indicating that the procedures for depositing AgNPs might not compromise the MNs’ strength.

To investigate the skin penetration capability, SMNs@PDA-AgNPs were punctured into both aluminum foil and excised rat skin. As shown in Figure 2f, 144 micropores (12 × 12) were created on the foil by SMNs@PDA-AgNPs, and the white paper was stained with trypan blue solution via these micropores. Similarly, there are 144 micropores on the skin surface, confirmed by trypan blue staining. Therefore, the PE of SMNs@PDA-AgNPs for both foil and skin is 100%.

The result of H&E staining is shown in Figure 2g. Two micropores created by the MNs were found in the skin tissue, with heights of 114.2 and 169.7 μm, respectively. The scale of the micropores is smaller than the size of the MNs because of the elasticity and irregular surface of the skin [34]. Nevertheless, it is clear that the MNs have penetrated both the stratum corneum and epidermis to reach the dermis layer. Notably, after insertion into both foil and skin, no obvious deformation or fracture of SMNs@PDA-AgNPs was observed. This may be due to the high hardness of the dry PVA sponge [35], which is the base material of the MNs. Collectively, the results suggest that SMNs@PDA-AgNPs have sufficient strength to penetrate the skin without breaking. 

### 3.4. Micromorphology and Chemical Composition 

The micromorphologies and chemical compositions of the various MNs were characterized by SEM and EDS. The results are shown in Figure 3a–f. SEM images in Figure 3a–d(i) showed that the four types of MNs had almost the same morphological characteristics. Specifically, they shared the same pyramidal shape for each needle tip (approximately 680 μm in height and 380 μm in base width) without any deformation or fracture, as shown in Figure 3a–d(ii). This indicates that the SMNs@PDA-AgNPs fabrication procedures may not significantly change the morphologies and structures. MNs at 680 μm in height were considered long enough to break the skin barrier for drug delivery and sampling, yet short enough to avoid stimulating nerves and causing pain [36]. The well-maintained morphology and structure of the MNs would ensure sufficient mechanical strength for effective skin penetration.

The surfaces of SMNs, SMNs@PDA, and SMNs-AgNO_3_ are shown in Figure 3a–c(iii), and they all had uneven, porous surfaces with no other nanoparticle aggregates attached. In contrast, some nanoparticle aggregates were found on the surface of SMNs@PDA-AgNPs, as shown in Figure 3d(iii). These substances may be AgNPs generated through in situ reduction by PDA. The presence of PDA coatings can be verified by comparing the surfaces of the different MNs. As shown in Figure 3a(iii),c(iii), the surfaces of SMNs and SMNs-AgNO_3_ without PDA coatings showed many tiny, cracked microstructures, which may reflect the microstructure of the sponge. However, no apparent cracked microstructures were observed on the surfaces of either SMNs@PDA or SMNs@PDA-AgNPs, as shown in Figure 3b(iii),d(iii), respectively. This may result from the PDA coatings, which are mussel-inspired membranes generated by the self-polymerization of DA [37].

The EDS patterns of these MNs are shown in Figure 3a–d(iv), respectively. There was no Ag detected in either SMNs or SMNs@PDA. However, a small amount of Ag (0.09%, summarized in the inset table) was found in SMNs-AgNO_3_, which may result from residual Ag ions adsorbed on the porous structure of the sponge. In contrast, large amounts of Ag (5.04%) were deposited on the surface of SMNs@PDA-AgNPs, confirming that the nanoparticle aggregates were AgNPs. The AgNPs were reduced in situ on the surface of SMNs@PDA-AgNPs without the introduction of any other reduction reagents because of the reduction capacity of catechol moieties on the PDA coatings. The PDA coatings could effectively reduce AgNO_3_ to generate AgNPs [38].

The morphology of AgNPs was examined by SEM, as shown in Figure 3e, indicating that the spherical AgNPs were uniform and appropriately 40 to 60 nm in diameter. In addition, the element mapping in Figure 3f showed a uniform distribution of elements (C, O, Si and Ag) in SMNs@PDA-AgNPs, further verifying the existence of AgNPs. Moreover, XRD patterns were recorded to finally confirm the formation of AgNPs. Figure 3g shows the XRD patterns of SMNs, SMNs@PDA, SMNs-AgNO_3_, and SMNs@PDA-AgNPs. In the XRD pattern of SMNs@PDA-AgNPs, diffraction peaks of 32.2° and 46.2° could be assigned to the crystal planes (122) and (200) of the face-centered cubic structure of Ag, respectively [22,39]. In contrast, no similar peaks were found in the XRD patterns of the other MNs, suggesting that AgNO_3_ could be successfully converted to AgNPs only in the presence of PDA coatings. Collectively, the results of SEM, EDS, and XRD show that AgNPs were successfully deposited on the surface of SMNs@PDA-AgNPs.

### 3.5. Antibacterial Activities of the MNs 

MNs-based medical devices for the long-term monitoring of chronic diseases are highly required to have antibacterial activities due to the risk of bacterial infection. To evaluate the antibacterial activities of these MNs, the MNs were inserted into the agar disc coated with *E. coli* for 24 h. To study the antibacterial activities of Ag^+^, SMNs-AgNO_3_ used in this test were not rinsed with water. As shown in Figure 4a(i), all MNs except SMNs (No. 1) have a clear and transparent ZOBI surrounding them, indicating that SMNs@PDA (No. 2), SMNs-AgNO_3_ (No. 3), and SMNs@PDA-AgNPs (No. 4) have antibacterial activities. The diameters of the ZOBI of Nos. 2, 3, and 4 are 22.4 ± 1.7, 28.2 ± 0.4, and 31.5 ± 1.4 mm, respectively, as shown in Figure 4a(ii). The diameter of the ZOBI of No.4 was significantly larger than that of both No.2 (*p* < 0.01) and No.3 (*p* < 0.05), suggesting that SMNs@PDA-AgNPs have better antibacterial activity than the other MNs. Both SMNs@PDA and SMNs-AgNO_3_ have the ability to inhibit bacteria due to the antibacterial activities of PDA and Ag^+^ [40,41]. The antibacterial activity of SMNs@PDA-AgNPs can be attributed mainly to the release of Ag^+^ from AgNPs. The antibacterial mechanism of the MNs is depicted in Figure 4a(iii). After SMNs@PDA-AgNPs were inserted into the agar disc, Ag^+^ was released from the MNs and could inhibit or kill bacteria through several mechanisms, including disruption of the cell wall, formation of free radicals, damage to DNA, and destabilization of ribosome [13]. The results show that SMNs@PDA-AgNPs have the potential for antibacterial application.

To optimize the antibacterial activity of SMNs@PDA-AgNPs, the key factors for preparing the MNs, including the concentration and incubation time of both DA and AgNO_3_ solutions as shown in Figure 4b, were further investigated. The results are shown in Figure 4c,d. When the DA concentration reached 0.4 mg/mL, the MNs had the highest antibacterial activity with the largest ZOBI diameter of 31.8 ± 1.4 mm. The activity of this concentration was significantly higher than the other concentrations except for 0.8 mg/mL (*p* > 0.05). The optimal incubation time for DA was found to be 2 h, as the best antibacterial activity (39.7 ± 0.4 mm) was achieved during this period, which was significantly higher than the other time periods (*p* < 0.001). Although the concentration of 1 mM AgNO_3_ resulted in the highest antibacterial activity with a ZOBI diameter of 22.4 ± 3.8 mm, there was no significant difference among the concentrations below 1 mM. However, the antibacterial activity of this concentration was significantly higher than that of 2 (*p* < 0.05) and 4 mM (*p* < 0.01). When the incubation time for AgNO_3_ was set at 4 h, the ZOBI diameter of 21.2 ± 0.3 mm was significantly larger than the other time periods, indicating that it had the best antibacterial activity. Therefore, the optimal reaction conditions in our laboratory are considered to be a DA concentration of 0.4 mg/mL and AgNO_3_ concentration of 1 mM, with incubation times of 2 and 4 h, respectively. In particular, the reaction conditions were determined based on the results of single-factor experiments. Response surface methodology will be explored to further optimize the reaction conditions due to the complex formation of both PDA and AgNPs [20].

### 3.6. Effects of MNs on Wound Healing

The wound healing experiment was conducted to verify the antibacterial activity of SMNs@PDA-AgNPs in vivo. The whole experiment was performed as shown in Figure 5a,b, and the results are shown in Figure 5c,d. An infected wound was constructed by allowing the growth of *E. coli* in the wound for 2 days, which would result in the formation of bacterial biofilms that would lead to delayed wound healing [26]. Biofilms are clusters of bacteria embedded in a self-produced matrix [42], which could be observed in all the groups, as shown in Figure 5d(i)–(iii) (0 day). Wound healing ratios in the SMNs group (i), SMNs@PDA-AgNPs group (ii), and SMNs + CIP group (iii) are all shown in Figure 5c. It was found that on the first 2 days, there was no significant difference among these groups (*p* > 0.05). However, after 4 days, the wound healing ratios of group (ii) and group (iii) were significantly higher than group (i) at each time point (*p* < 0.01). Specifically, the wound healing ratios of groups (i)–(iii) were 32.6 ± 5.7%, 52.4 ± 9.4%, and 51.5 ± 10.2% on day 4, respectively. 

As time went on, the wounds of different groups gradually healed. After 14 days, the wounds from the groups (ii) and (iii) were almost completely healed with healing ratios of 93.4 ± 3.2% and 88.0 ± 3.3%, respectively, which were both significant (*p* < 0.01) higher than group (i) (78.1 ± 4.3%). The results suggest that SMNs@PDA-AgNPs and SMNs + CIP promote the progress of wound healing. This can be attributed to the fact that MNs could puncture biofilms and release antibacterial agents (Ag^+^ or CIP) to inhibit or kill bacteria, which has been demonstrated by numerous previous studies [43]. Therefore, the antibacterial activity of SMNs@PDA-AgNPs was verified in vivo. 

### 3.7. Antibacterial Activity of MNs for Long-Term Application

Although the in vivo antibacterial activity of SMNs@PDA-AgNPs has been verified by the wound healing test, the antibacterial activity of MNs in practical application scenarios remains to be further confirmed. A novel method has been developed to study the antibacterial activity of MNs for long-term application. The method is shown in Figure 6a. The experiment was divided into the following three groups: (i) SMNs without bacterial infection; (ii) SMNs@PDA-AgNPs with bacterial infection; and (iii) SMNs with bacterial infection. The MNs were applied to the rats for 2 days and then removed. As shown in Figure 6b, after 2 days of MNs treatment, hundreds of micropores were developed on the skin of each group. It was found that the micropores created by group (iii) were more obvious and clearer than the other groups. Three days later, the micropores of groups (i) and (ii) were invisible to the naked eye, whereas several micropores could still be observed in group (iii). All the micropores in each group disappeared until 4 days later. 

The recovery process of micropores in the skin can be verified by changes in TEWL [44]. As shown in Figure 6c, all TEWL values on day 2 were significantly higher than on day 0, indicating that the skin barrier was damaged and micropores were created by MNs. Specifically, TEWL in groups (i) and (ii) on day 2 was 27.3 ± 1.5 and 23.4 ± 3.5 g/m^2^∙h, respectively, which were both significantly lower (*p* < 0.001) than in group (iii) (44.3 ± 6.1 g/m^2^∙h), suggesting that damage to the skin barrier by group (iii) was more severe than the others. On day 3, TEWL in all groups decreased dramatically, suggesting that micropores healed over time. However, the value of group (iii) (18.6 ± 0.7 g/m^2^∙h) was still significantly higher (*p* < 0.001) than the others. This further demonstrated that more severe skin damage was caused by group (iii), which took longer to recover from. By day 4, TEWL in all groups had returned to normal. The trend in TEWL changes was consistent with the micropores recovery process.

In addition, one rat from each group was sacrificed on day 2, and the treated skin was stained with H&E. The skin slices of each group are shown in Figure 6d, which further confirms that the micropores created by group (iii) were larger than those of the other groups. This suggested that bacterial infection delayed the healing process of micropores. Additionally, sebaceous gland hyperplasia below the micropores was apparently observed in group (iii), suggesting the presence of bacterial infection and chronic inflammation that stimulated the proliferation of sebaceous glands [45,46]. 

However, there was no significant difference in sebaceous glands between group (ii) and normal skin, indicating that SMNs@PDA-AgNPs inhibited or killed *E. coli* during application. In summary, all the results showed that SMNs@PDA-AgNPs had antibacterial activity for at least 2 days during application. It can be concluded that SMNs@PDA-AgNPs have the potential to be used as a component of medical devices or as a device for direct ISF sampling, as the antibacterial properties of MNs are essential to avoid the risk of bacterial infection [11]. 

### 3.8. Sampling ISF by MNs for Analysis with Safety 

As a component of medical devices, one of the most important properties of SMNs@PDA-AgNPs is the ability to sample ISF in vivo. The MNs procedures for sampling and detecting ISF are illustrated in Figure 7a. It is known that at least 1 μL (approximately 1 mg) of ISF should be collected, which is sufficient for multiple analytical assays [47]. Therefore, the MNs application time was set at 20 min, which ensures that the amount of ISF collected was more than 1 mg, as shown in Figure 7b. It was found that the amounts of ISF collected from the five rats were greater than 1.1 mg, and there was no significant difference among these rats (*p* > 0.05). The amounts of ISF were lower than the liquid extracted in vitro, which is consistent with our previous study [21]. Glucose levels in the blood and ISF are shown in Figure 7c. Glucose levels in each rat were significantly different (*p* < 0.01), which could be attributed to large individual differences in rats. Most of the glucose levels in blood and ISF from the same rat showed significant differences (*p* < 0.05), but the glucose levels in ISF were only slightly lower than in blood. This may be due to a time shift and a distorted mirror relationship between the glucose levels in blood and ISF, suggesting that a correction to the glucose level of ISF may be needed for diabetes management [48]. Similarly, cholesterol levels show a similar tendency in Figure 7d. In summary, it was found that both glucose and cholesterol levels determined from ISF samples were strongly correlated with blood sample values, indicating that the SMNs@PDA-AgNPs designed could be used for ISF sample analysis of biomarkers in vivo. Because of the absorption capacity of ISF, SMNs@PDA-AgNPs also have the potential to serve as a component of medical devices for real-time monitoring.

Moreover, after removal of the MNs, there were hundreds of micropores left on the skin surface, which could become channels for foreign agents or organisms (bacteria and viruses) to enter the body. Therefore, the recovery speed of these micropores is involved in the safe application of MNs. As shown in Figure 7e, after 20 min application of MNs, micropores were apparently observed on the skin that were accompanied by redness and swelling at 0 h. Half an hour later, the micropores could not be seen by the naked eye, and the redness and swelling of the skin almost completely disappeared. As time went on, the skin gradually returned to normal. This was further confirmed by the values of TEWL, which are described in Figure 7f. TEWL increased to 28.3 ± 0.5 g/m^2^∙h immediately after removal of MNs (0 h), which was significantly higher than the control (8.3 ± 0.2 g/m^2^∙h). This suggests that micropores were created by MNs and resulted in damage to the skin barrier. Over time, TEWL decreased gradually to 8.3 ± 0.05 g/m^2^∙h at 10 h, with no significant difference from the control. The results demonstrated that the micropores completely disappeared and the skin barrier recovered after removal of MNs for 10 h. Rapid recovery of skin barrier function reduces the potential risk of infection, indicating that SMNs@PDA-AgNPs are highly safe for ISF sampling [49]. 

## 4. Conclusions

In summary, SMNs@PDA-AgNPs were designed and developed to avoid bacterial infection during MNs application. AgNPs were successfully deposited on the surface of SMNs by in situ reduction of AgNO_3_ by PDA. Physiochemical characterization results indicate that MNs have sufficient mechanical strength to penetrate the skin and have the ability to extract sufficient liquid from an agar gel. The agar diffusion test verified and optimized the antibacterial activity of MNs in vitro. The wound healing assay proved that MNs could inhibit or kill bacteria in vivo and promote the healing process. Furthermore, the antibacterial activity of MNs was further confirmed during the application of MNs for 2 days. Finally, MNs were used to sample ISF for detecting glucose and cholesterol, the levels of which are closely correlated with those in blood. The developed SMNs@PDA-AgNPs show promising application prospects in MNs-based sampling and medical devices for biomarker monitoring. 

## Figures and Tables

**Figure 1 pharmaceutics-15-01730-f001:**
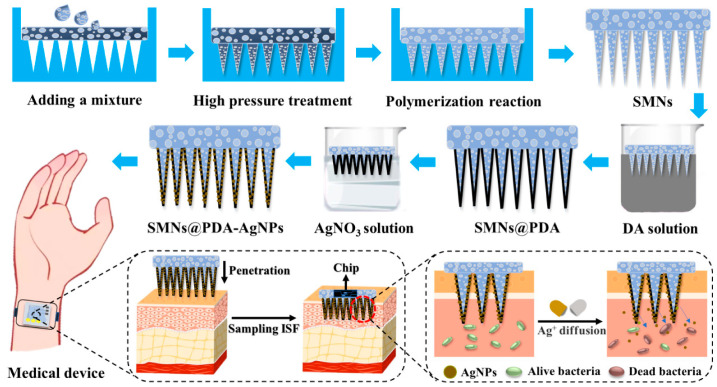
Scheme of the fabrication and application of SMNs@PDA-AgNPs for medical devices.

**Figure 2 pharmaceutics-15-01730-f002:**
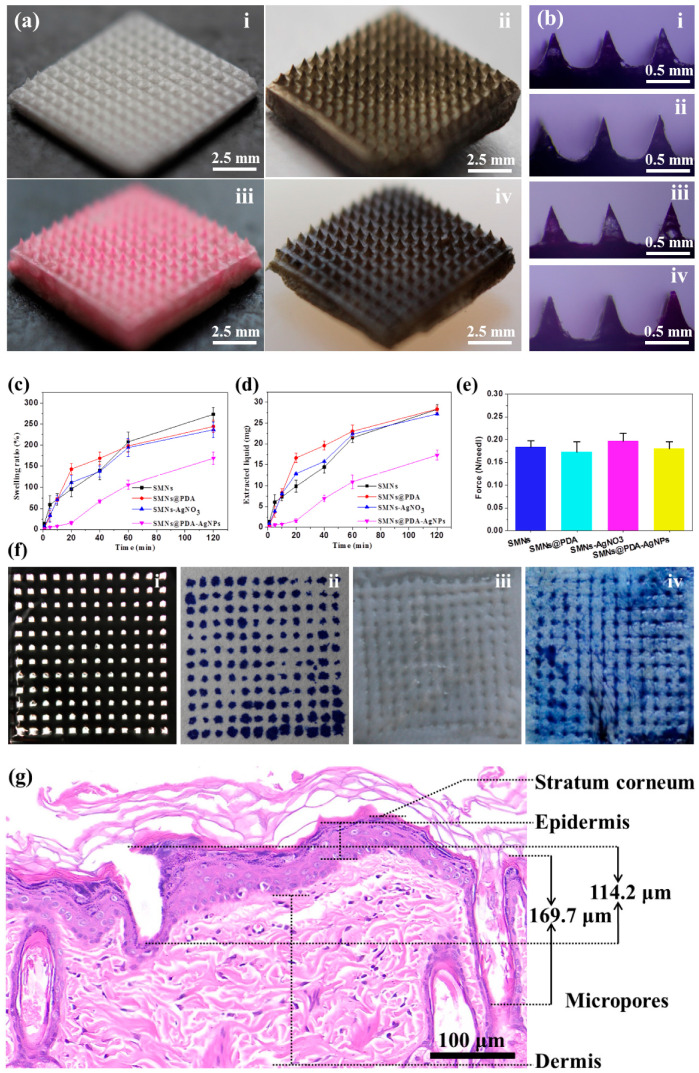
Macroscopic characterization of MNs: (**a**,**b**) photographs and optical microscopic pictures of (**i**) SMNs, (**ii**) SMNs@PDA, (**iii**) SMNs-AgNO_3_, and (**iv**) SMNs@PDA-AgNPs; (**c**) swelling ratios of the MNs; (**d**) extracted liquid by the MNs; (**e**) mechanical strength of the MNsl; (**f**) aluminum foil (**i**) and skin (**iii**) after puncture by SMNs@PDA-AgNPs, as well as trypan blue stained white paper (**ii**) and skin (**iv**); (**g**) H&E staining of skin treated with the MNs.

**Figure 3 pharmaceutics-15-01730-f003:**
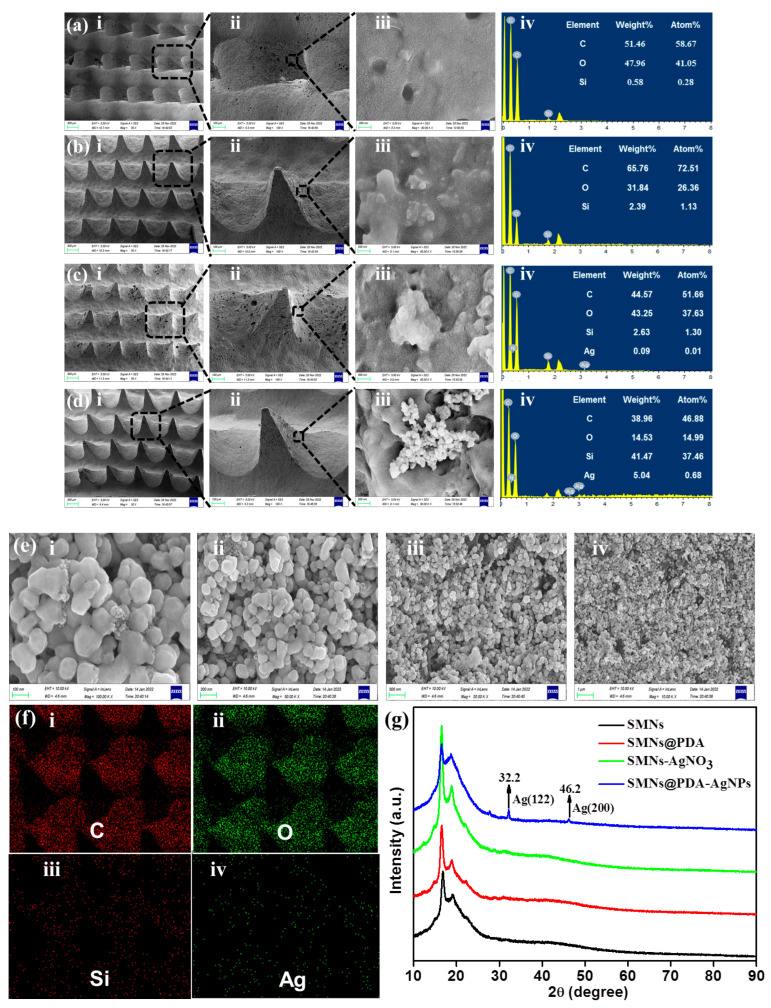
Micromorphology and chemical composition of MNs. SEM images and EDS patterns of (**a**) SMNs, (**b**) SMNs@PDA, (**c**) SMNs-AgNO_3_, and (**d**) SMNs@PDA-AgNPs: (**i**) top view of the micro-needle array, (**ii**) top view of an individual micro-needle, (**iii**) local surface morphology, and (**iv**) EDS pattern; (**e**) SEM images of AgNPs on the surface of SMNs@PDA-AgNPs at different magnifications: (**i**) 100 K X, (**ii**) 50 K X, (**iii**) 20 K X, and (**iv**) 10 K X; (**f**) EDS mapping of SMNs@PDA-AgNPs: (**i**) Carbon (C) element, (**ii**) Oxygen (O) element, (**iii**) Silicon (Si) element, and (**iv**) Silver (Ag) element; (**g**) XRD patterns of SMNs, SMNs@PDA, SMNs-AgNO_3_, and SMNs@PDA-AgNPs.

**Figure 4 pharmaceutics-15-01730-f004:**
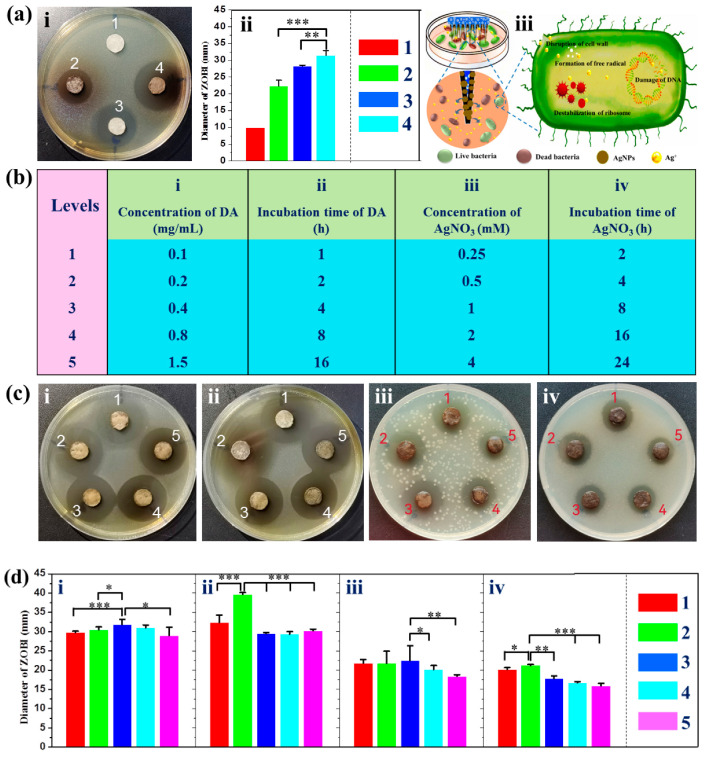
Antibacterial activities of MNs: (**a**) (**i**) photographs of the zone of bacterial inhibition (ZOBI) surrounding the MNs; (**ii**) ZOBI diameter of the MNs; (**iii**) scheme of the antibacterial mechanism by SMNs@PDA-AgNPs; (**b**) concentration and incubation time optimization: (**i**) concentration of DA, (**ii**) incubation time of DA, (**iii**) concentration of AgNO_3_, and (**iv**) incubation time of AgNO_3_.; (**c**,**d**) photographs and diameter of the ZOBI under the various reaction conditions: (**i**) concentration of DA, (**ii**) incubation time of DA, (**iii**) concentration of AgNO_3_, and (**iv**) incubation time of AgNO_3._ * *p* < 0.05, ** *p* < 0.01 and *** *p* < 0.001 are considered statistically significant. Each test was repeated at least three times (*n* = 3).

**Figure 5 pharmaceutics-15-01730-f005:**
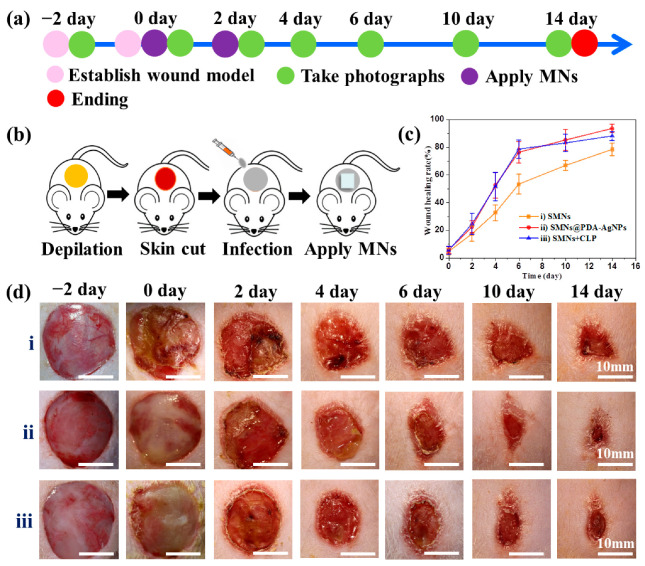
Effects of MNs on wound healing: (**a**,**b**) scheme of the experimental procedures for the application of MNs in wound healing; (**c**) wound healing rates of rats with different treatments; (**d**) Photographs of wound healing from day −2 to day 14 with different treatments: SMNs group (**i**), SMNs@PDA-AgNPs group (**ii**), and SMNs + CIP group (**iii**). Data are presented as mean ± SD (*n* = 5).

**Figure 6 pharmaceutics-15-01730-f006:**
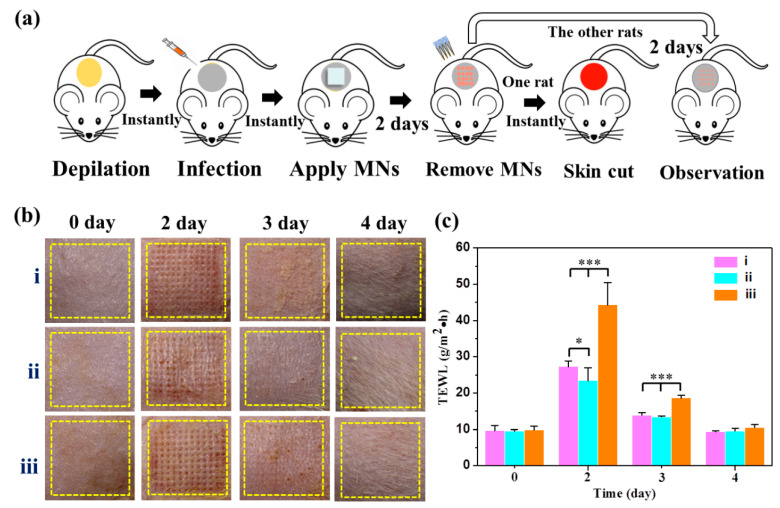
Antibacterial activity of MNs during application: (**a**) scheme of the experimental procedures; (**b**) recovery process of micropores created by MNs: (**i**) SMNs without bacterial infection, (**ii**) SMNs@PDA-AgNPs with bacterial infection, and (**iii**) SMNs with bacterial infection; (**c**) trends in changes of TEWL; (**d**) histological view of skin stained with H&E. Data are presented as mean ± SD (*n* = 5). * *p* < 0.05 and *** *p* < 0.001 are considered statistically significant.

**Figure 7 pharmaceutics-15-01730-f007:**
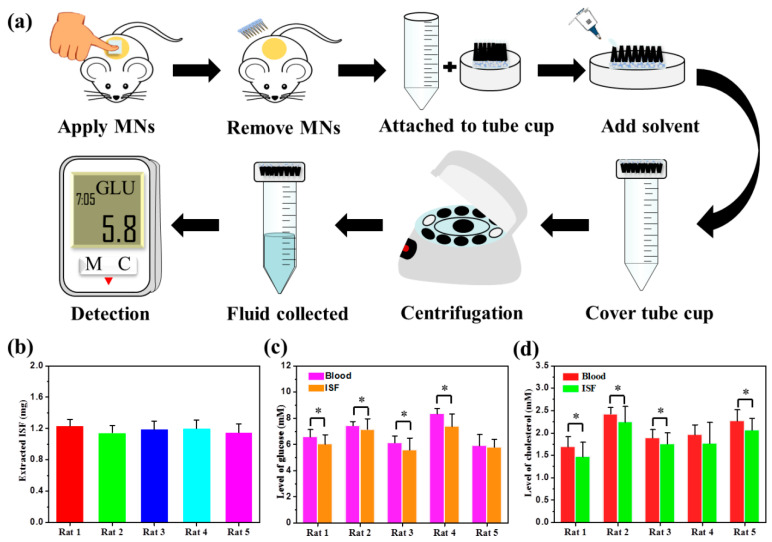
Sampling of ISF by MNs for safety analysis: (**a**) scheme of the ISF sampling procedures using MNs for analysis; (**b**) the amounts of ISF extracted from five individual rats; (**c**,**d**) comparison of glucose (**c**) and cholesterol (**d**) levels in blood and ISF; (**e**) the recovery of skin treated with MNs; (**f**) changes in TEWL values after MNs treatment. * *p* < 0.05 is considered statistically significant.

## Data Availability

The data underlying this article will be shared on reasonable request by the corresponding author.

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
