# Peer review of "Fabrication of Antibacterial Sponge Microneedles for Sampling Skin Interstitial Fluid"

_pharmaceutics, 2023, doi:10.3390/pharmaceutics15061730_

Round 1

Reviewer 1 Report

The article is well structured and documented, being in line with the guidelines for the authors, imposed by the journal.

The abstract is well structured and clearly presented, according to this research.

The state-of-the-art presented in Introduction of the paper is well documented and focused on the actual research direction in the field, underlining the advantages regarding of antibacterial sponge microneedles by depositing silver nanoparticles on polydopamine coated sponge microneedles. The ability to extract sufficient liquid from an agar gel and the antibacterial activity were demonstrated.

The experimental procedure is well justified and comprehensibly showed.

The results are presented in a concise manner and sustained by proper figures and tables. Furthermore, the statistical analysis indicates that the results obtained are statistically significant.

The discussion section is clear, understandable and in accordance with the results obtained.

The conclusion is well explained by the results of the experiments performed.

The references indicate a very well documentation. 

Observations

Some very minor corrections are required - e.g blank spaces L 43, 130

Author Response

A:Thank you for acknowledging and expressing gratitude for the positive assessment provided by you. Your feedback is valuable and serves as validation for the quality and contribution of our article. We are grateful for your recognition and support, which further motivates us in our research endeavors.

A1:We have conducted a comprehensive review of the entire manuscript once again, aiming to identify and rectify any minor errors. We sincerely appreciate your suggestions and guidance, which have prompted us to carefully scrutinize the paper. By taking this diligent approach, we aim to ensure the accuracy and integrity of the content. We are grateful for your valuable input and will continue to strive for excellence in our work. Thank you for your continued support and guidance.

Reviewer 2 Report

Manuscript Fabrication of antibacterial sponge microneedles for sampling skin interstitial fluid” is interesting research for ISF extraction and design of new medical devices. Authors developed novel antibacterial sponge SMNs@PDA-AgNPs by depositing AgNPs on polydopamine (PDA)-coated SMNs. This work can be published in Pharmaceutics after some corrections.

1. It is desirable to describe in more detail the methods of obtaining SMNs@PDA-AgNPs in section 2.4. Dopamine in Tris buffer (pH 8.5) readily oxidizes to form quinones. How was the quality of dopamine controlled at this stage? Are quinones formed as a by-product? What characteristics were used to evaluate the quality of the dopamine self-polymerization product?

2. There is a problem of AgNPs aging. Did silver oxides Ag2O appear on the surface of SMNs@PDA-AgNPs? Did aggregation or agglomeration proceed during the storage of SMNs@PDA-AgNPs? Perhaps the antibacterial effect is also due to the additional presence of Ag2O and Ag+ ions.

3. Figure 3g. Please provide zoomed-in XRD patterns in the range of 25-50 degrees with increased signal intensity. Now, the intensity of the Ag(122) and Ag(200) reflections is very low.

Author Response

A:Thank you for your positive feedback on the manuscript "Fabrication of antibacterial sponge microneedles for sampling skin interstitial fluid." We appreciate your recognition of its significance in the field of interstitial fluid extraction and the design of new medical devices. We have carefully and thoroughly revised the manuscript based on your comments, hoping to meet the requirements.

A1:Thank you for your valuable advice. The quality of dopamine was effectively controlled through the regulation of its concentration, as highlighted in section 2.4. Under alkaline conditions, dopamine undergoes oxidation, resulting in the formation of quinones. It is worth noting that quinones play a significant role as intermediate products, rather than being considered by-products. Characterizing the self-polymerization of dopamine proved to be a challenging task. However, the formation of polydopamine (PDA) was successfully achieved by precisely controlling the concentration of dopamine and the reaction time. To further elaborate, we have incorporated additional details regarding the formation of PDA and its role in facilitating the reduction of silver nanoparticles in the revised section "3.1. Fabrication and macromorphology of MNs." This addition provides a more comprehensive understanding of the mechanism behind PDA-assisted silver nanoparticle synthesis.

A2:Your questions are highly pertinent, especially regarding the issue of silver nanoparticle aging. It is indeed an area that we did not specifically investigate in our study. While we have not studied the oxidation of silver nanoparticles into silver oxides or the occurrence of aggregation during storage, your inquiries raise important considerations for further investigation. The antibacterial effect observed might indeed be attributed to the additional presence of Ag2O and Ag+ ions. However, it is worth mentioning that we have taken measures to minimize the influence of silver ions through repeated washing and soaking, ensuring that any antibacterial effects are primarily attributed to the presence of silver nanoparticles. Your comments have provided valuable input for our future work, and we appreciate your insightful perspective.

A3:Thank you for providing the additional information. We apologize for any confusion caused. We understand that you are requesting zoomed-in XRD patterns in the range of 25-50 degrees with increased signal intensity for the Ag(122) and Ag(200) reflections in Figure 3g. However, we must clarify that the ability to enhance the signal intensity of the XRD patterns or modify the original data is beyond the scope of our study. The current low intensity of the Ag(122) and Ag(200) reflections may be attributed to the specific region chosen for sample analysis during XRD measurements. We appreciate your feedback and understanding regarding the limitations in modifying the XRD data. If there are any further clarifications or additional analyses you would like us to provide, please let us know, and we will do our best to assist you.

Reviewer 3 Report

The manuscript is well-written/organized. I just have a few comments before its acceptance to Pharmaceutics.

1)      What was the size of Ag NPs in the SMNs@PDA-AgNPs sample?

2)      A mechanism to show the PDA-assisted Ag NPs synthesis should be given.

3)      Please provide UV-vis spectra of the SMNs@PDA-AgNPs.

4)      Some latest studies on Ag-based antibacterial/wound dressing materials should be cited. One example:

-       https://doi.org/10.3390/polym11071185

Author Response

A1:Thank you for your careful review and questions. The size of AgNPs in the SMNs@PDA-AgNPs sample was found to be approximately 40 to 60 nm, as determined through SEM analysis. This information was also presented in the "3.4. Micromorphology and chemical composition" section. The sentence is as follow:“The morphology of AgNPs was examined by SEM, as shown in Figure 3e, indicating that the spherical AgNPs were uniform and appropriately 40 to 60 nm in diameter.”

A2:According to your suggestion, I have made some revisions to the text in section "3.1. Fabrication and macromorphology of MNs." Here are the revised sentences: "The mechanism of PDA-assisted synthesis of AgNPs can be summarized as follows: Dopamine undergoes a series of reactions, ultimately resulting in the formation of PDA. PDA is characterized by active phenolic hydroxyl groups, which enable effective interaction with silver ions, leading to their reduction and subsequent synthesis of AgNPs."

A3:We would like to express our sincere appreciation to the reviewer for their valuable comment. However, according to the nature of SMNs@PDA-AgNPs being solid microneedles with silver nanoparticles attached in a particulate form, it is not possible to obtain UV-vis spectra since it is not in a solution state.

A4:According to your suggestion, the following three recent research papers have been cited. [1]. Pant, B.; Park, M.; Park, S.-J. One-step synthesis of silver nanoparticles embedded polyurethane nano-fiber/net structured membrane as an effective antibacterial medium. Polymers, 2019, 11 (7), 1185. [2].LewisOscar, F.; Nithya, C.; Vismaya, S.; Arunkumar, M.; Pugazhendhi, A.; Nguyen-Tri, P.; Alharbi, S. A.; Alharbi, N. S.; Thajuddin, N. In vitro analysis of green fabricated silver nanoparticles (AgNPs) against Pseudomonas aeruginosa PA14 biofilm formation, their application on urinary catheter. Progress in Organic Coatings, 2021, 151, 106058. [3]. Lara, H. H.; Ixtepan-Turrent, L.; Jose Yacaman, M.; Lopez-Ribot, J. Inhibition of Candida auris biofilm formation on medical and environmental surfaces by silver nanoparticles. ACS applied materials & interfaces, 2020, 12 (19), 21183-21191.

Round 2

Reviewer 2 Report

Dear Authors!

Thank you for answers and corrections of manuscript.

You have addressed my concerns with the original manuscript.